# A Support Vector Machine Forecasting Model for Typhoon Flood Inundation Mapping and Early Flood Warning Systems

**Ming-Jui Chang [1], Hsiang-Kuan Chang [1], Yun-Chun Chen [2], Gwo-Fong Lin [3,*] , Peng-An Chen [4], Jihn-Sung Lai [1,2,4,5] and Yih-Chi Tan [1,2,5]**

[1] Research Center of Climate Change and Sustainable Development, National Taiwan University, Taipei 10617, Taiwan; f99521311@ntu.edu.tw (M.-J.C.); akuanchang@gmail.com (H.-K.C.); jslai525@ntu.edu.tw (J.-S.L.); yctan58@gmail.com (Y.-C.T.)
[2] Department of Bioenvironmental Systems Engineering, National Taiwan University, Taipei 10617, Taiwan; a22802901@gmail.com
[3] Department of Civil Engineering, National Taiwan University, Taipei 10617, Taiwan
[4] Hydrotech Research Institute, National Taiwan University, Taipei 10617, Taiwan; a0932195109@yahoo.com.tw
[5] Center for Weather Climate and Disaster Research, National Taiwan University, Taipei 10617, Taiwan
* Correspondence: gflin@ntu.edu.tw; Tel.: +886-2-33664368; Fax: +886-2-23631558

**Abstract:** Accurate real-time forecasts of inundation depth and extent during typhoon flooding are crucial to disaster emergency response. To manage disaster risk, the development of a flood inundation forecasting model has been recognized as essential. In this paper, a forecasting model by integrating a hydrodynamic model, *k*-means clustering algorithm and support vector machines (SVM) is proposed. The task of this study is divided into four parts. First, the SOBEK model is used in simulating inundation hydrodynamics. Second, the *k*-means clustering algorithm classifies flood inundation data and identifies the dominant clusters of flood gauging stations. Third, SVM yields water level forecasts with 1–3 h lead time. Finally, a spatial expansion module produces flood inundation maps, based on forecasted information from flood gauging stations and consideration of flood causative factors. To demonstrate the effectiveness of the proposed forecasting model, we present an application to the Yilan River basin, Taiwan. The forecasting results indicate that the simulated water level forecasts from the point forecasting module are in good agreement with the observed data, and the proposed model yields the accurate flood inundation maps for 1–3 h lead time. These results indicate that the proposed model accurately forecasts not only flood inundation depth but also inundation extent. This flood inundation forecasting model is expected to be useful in providing early flood warning information for disaster emergency response.

**Keywords:** early flood warning; disaster risk; *k*-means clustering algorithm; support vector machine; flood inundation forecasting; flood inundation map

## 1. Introduction

According to records from the past 100 years from the Central Weather Bureau, Taiwan, an average of three typhoons strike Taiwan each year [1]. During typhoons, heavy rainfall and widespread flood inundation typically impact the Island of Taiwan. To illustrate the extremity of these events, the average annual rainfall in Taiwan is around 2500 mm, with more than 78% of the precipitation concentrated in the typhoon season from May to October [2]. The torrential rainfall brought by typhoons frequently causes flood inundation, which leads to the loss of life and property. For disaster mitigation in the

form of an early flood warning system, the development of a flood inundation forecasting model has been recognized as a crucial task. A forecasting model can drastically improve the lead time of decision-making and assist the personnel in handling the emergency response [3]. However, the development of flood inundation forecasting models is always a challenge due to the complex nature of flooding processes and the typical lack of detailed hydro-geographical information in a given study area. Due to advances in numerical modeling techniques and computing capabilities, distributed mathematical and numerical models have recently become increasingly attractive solutions for hydrodynamic simulation. Examples of such models include the SOBEK model [4–6], FLO-2D [7], ANUGA Hydro [8], 2D zero-inertia inundation model [9] and 2D-DOFM (2D diffusive overland flow model) [10–12]. Of these approaches, the SOBEK model is a fully integrated modeling framework for river, estuary, and storm water systems, capable of simulating hydrodynamics of flood inundation phenomena. Doong et al. [13] indicated that the SOBEK model could simulate flood inundation depth during typhoons while taking into account tidal effects. Kuntiyawichai et al. [14] estimated flooding with a process-based hydrological model coupled to a 1D/2D SOBEK model for the estimation of flood retardation and damage mitigation. Verwey et al. [15] presented a conceptualization of the rainfall-runoff process using SOBEK for an urbanized catchment in Singapore. However, the SOBEK model requires various types of hydro-geomorphological information as well as long computation times to model flood inundation depths and extents. This again indicates that accurate flood inundation forecasting is difficult to achieve. Moreover, the uncertainty in physically based models also cause the difficulty of the flood inundation forecasting [16]. To obtain the real-time and accurate flood inundation forecasts, the artificial neural network (ANN), which provides rapid, precise and inexpensive estimates of network structures, is substituted for the physically based models [17].

ANN provides great flexibility in solving nonlinear problems and it has been applied in various aspects of hydrology [18–21]. Due to its superior ability in modeling nonlinear systems and its computational efficiency, ANN has also been employed for flood inundation forecasting [22–24]. ANN models, which include back propagation networks and support vector machines (SVM), etc., perform more efficiently for hydrodynamic forecasting than physically based models [25,26]. Among ANN models, SVM have been particularly effective in flood inundation forecasting modeling. Lin et al. [27] proposed a flood inundation forecasting model using SVM to yield 1–3 h lead time flood inundation maps. Lin et al. [27] defined control points to forecast the flood inundation map in their study. However, the location of a given control point was not the location of a flood gauging station, so the flood inundation data could not be directly obtained. To address this problem, flood inundation depths at flood gauging stations, rather than at control points, were employed as input in constructing the flood inundation forecasting model. Then, the flood causative factors are selected as candidate input in constructing the spatial expansion module. With the spatial expansion module, the flood inundation forecasts are expanded from points to areas and yield the flood inundation maps. The high accuracy and efficiency SVM in this study showed it to be suitable for rapidly producing forecasted flood inundation maps with real-time data and integrating with disaster warning systems.

Determination of the critical flood causative factors is crucial in the creation of useful flood inundation maps [28–32]. Flood causative factors can be commonly divided into topographic, geologic, geographical and location factors. However, typically not all candidate factors are selected as input for the model, since some factors may effectively act as noise. Hong et al. [28] considered eleven flood-related variables as input into the fuzzy WofE-SVM model, namely: lithology, soil cover, elevation, slope angle, aspect, topographic wetness index (TWI), stream power index (SPI), sediment transport index, plan curvature, profile curvature, and distance from river network. Using the ensemble technique of logistic regression and weight of evidence, Tehrany et al. [32] selected 15 flood causative factors as input for evaluating predictive performance in flood susceptibility mapping in China. The flood causative factors were altitude, slope, aspect, geology, distance from river, distance from road, distance from fault, soil type, land use/cover, rainfall, normalized difference vegetation index, SPI, TWI, sediment transport index and curvature. Fourteen factors potentially responsible for flooding

were determined and selected as input in a hybrid model by integrating the principal component analysis, logistic regression and frequency distribution analysis to quantify hazard potential and to map flood characteristics [30]. Among these studies, the flood causative factors are determined by using the correlation coefficient analysis. Nine flood causative factors are identified and adopted as input to the proposed model, including elevation, slope, aspect, curvature, plan curvature, profile curvature, TWI, distance to river, and SPI.

In the present study, flood inundation depths are simulated by the SOBEK model, as calibrated and validated by survey data of flood inundation extents and flood water levels. The flood inundation data are grouped into several clusters by a *k*-means clustering algorithm, in order to identify the dominant clusters for each flood gauging station. Then, rainfall and water level data are determined as input in developing the point forecasting module. Finally, based on rainfall data, point forecasts and flood causative factors, the spatial expansion module is constructed to expand the flood inundation depth from points to areas as flood inundation extents. To demonstrate the effectiveness of the flood inundation forecasting model, we explore an application to the Yilan River basin in Yilan County, Taiwan. Our study is organized as follows. The study area and data collection are described in Section 2. Section 3 presents the development of the flood inundation forecasting model, including hydrodynamic simulation, classification, point forecasting, and spatial expansion steps. The results and performance of the proposed model are presented and discussed in Section 4. The conclusions are summarized in Section 5.

## 2. Study Area and Hydrological Data

The Yilan River basin is located in northeastern Taiwan. The annual average precipitation and temperature are 2522 mm and 22 °C, respectively. The mainstream length is approximately 17.25 km, while the total watershed area is around 149.06 km$^2$. Several irrigation and drainage systems have been built in the Yilan River basin. The Meifu drainage system, located on the southern side of the Yilan River, is one of the major drainage systems. Typhoons usually hit this region in the summer and fall, from August to October. During typhoons, severe flood inundations may quickly form in low-lying areas between the Meifu drainage system and the Yilan River, causing serious property loss and damage.

The locations of the rainfall, water level and flood gauging stations in the Yilan River basin are shown in Figure 1. The total average hourly rainfall is calculated with Thiessen's Polygon method. Among the Thiessen area, the maximum area is dominated by Dajiaoxi rainfall station. It indicates that the rainfall at the Dajiaoxi rainfall station is crucial to affect the forecasting result. Table 1 presents hydrological data from ten typhoons, including duration, cumulative maximum within 36 h and flood inundation extents. The eight flood causative factors were calculated from the digital elevation model (DEM) data with a spatial resolution of 40 m × 40 m, consisting of elevation, slope, aspect, curvature, plan curvature, profile curvature, distance to river, SPI and TWI. The SPI and TWI are expressed as $A \times \tan(B)$ and $\ln(A/\tan(B))$, respectively, where $A$ is the upstream contributing area (m$^2$) and $B$ is the local slope (degree).

**Table 1.** Hydrological and inundation data for typhoons from 2004 to 2015.

| Event | Date (yyyy/mm/dd) | Duration (h) | Cumulative Maximum within 36 h Rainfall (mm) | Inundation Extent (km$^2$) |
|---|---|---|---|---|
| Aere | 2004/08/23 | 80 | 350.4 | 0.82 |
| Nanmadol | 2004/12/03 | 35 | 300.4 | 1.36 |
| Sepat | 2007/08/16 | 77 | 186.6 | 1.75 |
| Sinlaku | 2008/09/11 | 125 | 454.1 | 7.33 |
| Jangmi | 2008/09/27 | 71 | 185.6 | 0.10 |
| Parma | 2009/10/03 | 83 | 305.6 | 5.51 |
| Megi | 2010/10/21 | 68 | 416.0 | 6.35 |
| Saola | 2012/07/30 | 89 | 379.8 | 5.22 |
| Soudelor | 2015/08/06 | 80 | 214.9 | 0.87 |
| Dujuan | 2015/09/27 | 56 | 179.9 | 0.54 |

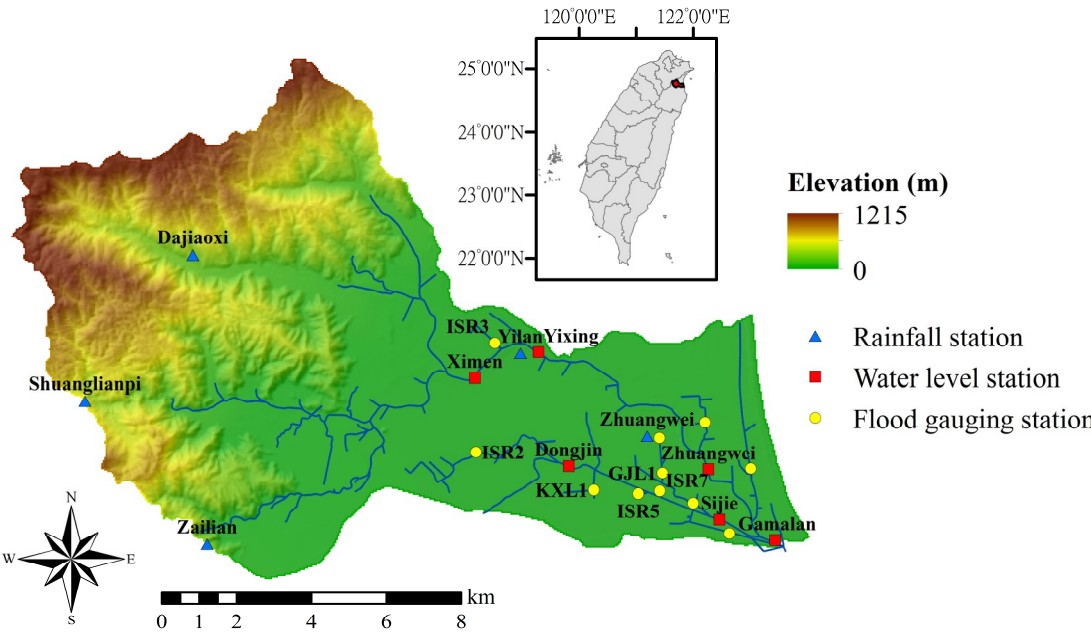

**Figure 1.** The Yilan River basin and the locations of rainfall, water level and flood gauging stations.

## 3. Methodology Development

### 3.1. Hydrodynamic Simulation

For hydrologic and hydraulic analysis, the SOBEK model was adopted for solving the Saint-Vernant equations [4]. The SOBEK model contains three major modules: a rainfall-runoff module (RR), a 1D flow module (1DF) and a 2D overland flow module (OF). The SOBEK model's finite difference scheme is commonly applied in simulating the unsteady flow velocities, water levels, and inundation extents associated with flooding events, both in rivers and in urban sewer/drainage systems.

Since the study area contains many drainage systems, we divide the basin into rural river areas, urban sewer areas and surface runoff catchments, each of which is simulated by a different analysis module. For rural river areas, the Soil Conservation Service (SCS) curve number is used to analyze the runoff for different rainfall events according to land use. For urban sewer areas, a rational formula is used to calculate the surface runoff discharge. The RR module with rainfall data calculates surface runoff discharge hydrographs.

For the 1D flow module, the continuity equation and the momentum equation are the dominant equations and can be expressed as

$$\frac{\partial A_f}{\partial t} + \frac{\partial Q}{\partial x} = q_{lat} \tag{1}$$

$$\frac{\partial Q}{\partial t} + \frac{\partial}{\partial x}\left[\frac{Q^2}{A_f}\right] + gA_f\frac{\partial h}{\partial x} + \frac{gQ|Q|}{C^2RA_f} - W_f\frac{\tau_{wi}}{\rho_w} = 0 \tag{2}$$

where $Q$ is the discharge (m³/s), $g$ is the acceleration due to gravity (m/s²), $t$ is the time (s), $h$ is the water level (m), $R$ is the hydraulic radius (m), $q_{lat}$ presents the lateral discharge per unit length (m²/s), $A_f$ is the cross sectional flow area (m²), $C$ is the Chezy coefficient, $W_f$ is the cross sectional width at the corresponding water level (m²), $\tau_{wi}$ is the wind shear stress ($\frac{\text{kg}}{\text{m}\times\text{s}^2}$), and $\rho_w$ is the water density (kg/m³).

The 2D overland flow module is described by the continuity equation and the momentum equation in the x- and y-directions. The continuity equation and momentum equations are expressed as

$$\frac{\partial h}{\partial t} + \frac{\partial (ud)}{\partial x} + \frac{\partial (vd)}{\partial y} = 0 \tag{3}$$

$$\frac{\partial u}{\partial t} + u\frac{\partial u}{\partial x} + v\frac{\partial u}{\partial y} + g\frac{\partial h}{\partial x} + g\frac{u|V|}{c^2 d} + au|u| = 0 \tag{4}$$

$$\frac{\partial v}{\partial t} + u\frac{\partial v}{\partial x} + v\frac{\partial v}{\partial y} + g\frac{\partial h}{\partial y} + g\frac{v|V|}{c^2 d} + av|v| = 0 \tag{5}$$

where $d$ is the water depth (m); $u$ and $v$ are the velocity components (m/s) in the x-direction and y-direction, respectively; $V$ is the velocity magnitude $\sqrt{u^2 + v^2}$ (m/s); $c$ is the Chezy coefficient; and $a$ is the wall friction coefficient.

The flow chart of the linkage of the integrated model is shown in Figure 2. The 1DF module calculates the hydrographs of overflow flow rate at channel cross-sections when the surface runoff exceeds the design capacity of the drainage system. The overflow hydrographs are subsequently used as sources in the OF module. Historical typhoon event records are used in calibrating some parameters: the SCS curve number (*CN*) and White Colebrook (*k*). The value of the Chezy coefficient $c$ is obtained from the White-Colebrook formula, expressed as $c = 18^{10}\log(\frac{12R}{k_n})$.

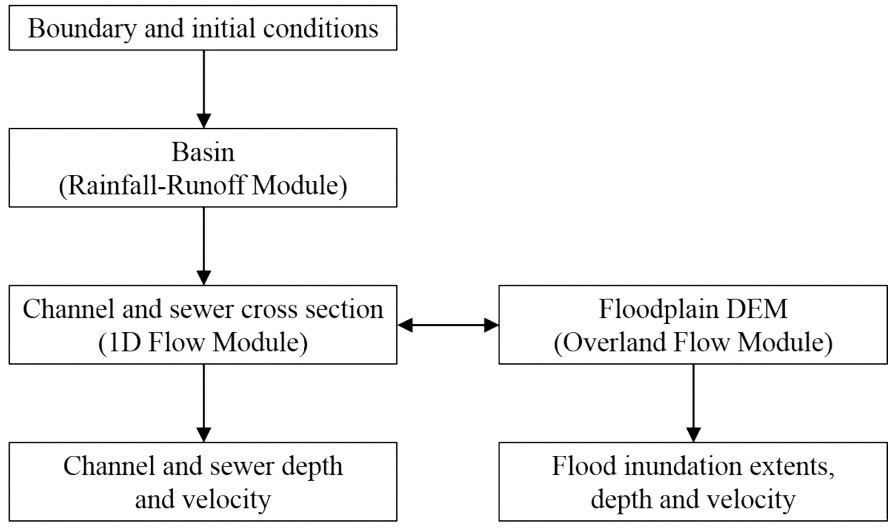

**Figure 2.** Flowchart of the SOBEK model.

*3.2. k-means Clustering Algorithm*

The *k*-means clustering algorithm [33] groups data with simulated statistical characteristics. Due to the ease of implementation and efficiency, the *k*-means algorithm is one of the most commonly used partitional clustering algorithms [34,35], and many studies have confirmed its clustering ability in hydrology [27,36]. Data are grouped into $n$ clusters by their common characteristics. The clusters are recognized by minimizing the average Euclidean distance from individual data points to the cluster center. The *k*-means clustering algorithm function is written as

$$E = \sum_{k=1}^{n} E_k = \sum_{i=1}^{n}\sum_{j=1}^{l} w_{ji}\left\|x_j - c_i\right\|^2 \tag{6}$$

where $n$ is the number of clusters, $x_j$ is the input vector, $c_i$ is the $i$th cluster center and $w_{ji}$ is a $l \times k$ data matrix. For more details on the *k*-means clustering algorithm, refer to [37].

In this study, similar flood inundation data are grouped into the same cluster. Then, clusters of flood inundation data are identified by minimizing their average Euclidean distance to the cluster center. Determination of an optimal number of clusters is crucial for efficiently grouping the flood inundation depths. In order to obtain an optimal number of clusters, the grid-search method [38] is used for determining the optimal number of clusters. Integrating a classification scheme like *k*-means into an SVM model is expected to reduce the number of the constructed models and increase the forecast accuracy.

### 3.3. Support Vector Machine

Vapnik [39] developed SVM for classification and later extended the technique for regression analysis. Classifier functions are used for separating different groups of training data to construct hyperplanes in the multidimensional space. In the past two decades, SVM has been widely adopted in rainfall-runoff [40,41], flood [27,42–44] and peak-flood analyses [45]. For further details on SVM, refer to [39,46].

A SVM model is a given training set $N_d[(x_1, y_1), (x_2, y_2), \cdots, (x_{N_d}, y_{N_d})]$, with input vector $x_i$ and output data $y_i$. The objective of the SVM is to find a nonlinear regression function $\hat{y} = f(x) = \mathbf{w}^T \phi(x) + b$ and produce the output $\hat{y}$, which is the optimal approximate of the observed data with an error tolerance of $\varepsilon$, where $\phi(x)$ is a nonlinear function, $\mathbf{w}$ is weight, and $b$ is the bias of the regression function, respectively. According to the structural risk minimization (SRM) induction principle, $\mathbf{w}$ and $b$ are calculated by minimizing the structural risk function:

$$R = \frac{1}{2}\|\mathbf{w}\|^2 + C\sum_{i=1}^{N_d} L_\varepsilon(\hat{y}) \tag{7}$$

where Vapnik's $\varepsilon$-insensitive loss function is defined as

$$L_\varepsilon(\hat{y}) = |y - f(x)|_\varepsilon = \begin{cases} 0 & \text{if}|y - f(x)| < \varepsilon \\ |y - f(x)| - \varepsilon & \text{if}|y - f(x)| \geq \varepsilon \end{cases} \tag{8}$$

The first term in Equation (2) represents model complexity; the second term represents empirical error. The penalty parameter $C_P$ represents the tradeoff between model complexity and empirical error.

The SVM problem can be formulated as the following optimization problem:

$$\min R(w, b, \xi, \xi') = \frac{1}{2}\|W\|^2 + C_P \sum_{i=1}^{N_d} (\xi_i + \xi'_i)$$

subject to

$$
\begin{aligned}
y_i - \hat{y}_i = y_i - (\mathbf{w}^T\phi(x_i) + b) &\leq \varepsilon + \xi_i \\
\hat{y}_i - y_i = (\mathbf{w}^T\phi(x_i) + b) - y_i &\leq \varepsilon + \xi'_i \\
\xi_i \geq 0, i = 1, 2, \cdots, N_d \\
\xi'_i \geq 0, i = 1, 2, \cdots, N_d
\end{aligned}
\tag{9}
$$

where $\xi$ and $\xi'$ are slack variables representing, respectively, the upper and lower training errors subject to error tolerance $\varepsilon$. The dual Lagrange multipliers, $a$ and $a'$, can be applied to solve the above optimization problem. Consequently, the approximate function can be expressed as follows:

$$f(x) = \sum_{i=1}^{m} (a_i - a'_i)K(x_i, x) + b \tag{10}$$

where *m* is the number of support vectors, $x_i$ is the support vector, and $K(x_i,x)$ is a kernel function, used for mapping the SVM input vector into a higher-dimensional feature space. The radial basis function (RBF) is used herein and expressed as follows:

$$K(x_i, x_j) = (-\gamma \|x_i - x_j\|), \gamma > 0 \tag{11}$$

where $\gamma$ is the gamma term. The correct determination of parameters significantly increases the accuracy of SVM solutions. The penalty parameter $C_\mathrm{p}$ and the error tolerance $\varepsilon$ are also crucial SVM parameters. In this study, the grid-search method is employed for determining the optimal combination of the kernel function, $\gamma$, $C_\mathrm{p}$ and $\varepsilon$. In this study, the SVM is applied to develop the point forecasting module to forecast the flood inundation depth for each flood gauging station. Then, the SVM is used to develop the spatial expansion module to expand the flood inundation depth from points to areas for each cluster.

### 3.4. Methodology Construction

To yield flood inundation maps in a time series, a flood inundation forecasting model is developed in this study. Figure 3 shows a flowchart of the flood inundation forecasting model, which consists of four steps: hydrodynamic simulation, classification of flood inundation data, point forecasting for flood inundation depth and spatial expansion for flood inundation extent. Details of these four steps are described below.

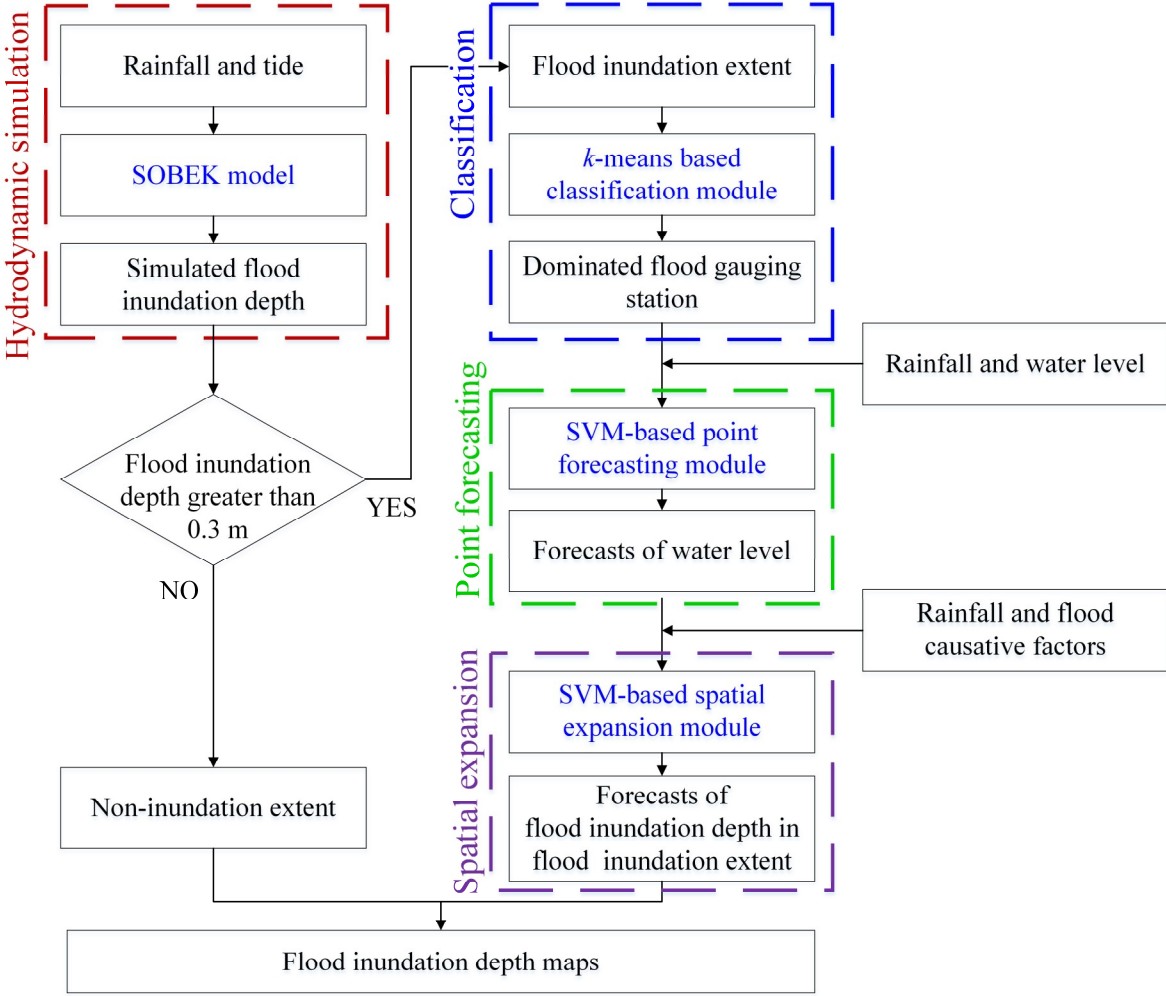

**Figure 3.** Flowchart of the proposed model.

### 3.4.1. Hydrodynamic simulation step

Flood inundation in the Yilan River basin is simulated using the SOBEK model. The river reach, cross section points of rivers and cross-sectional shapes of rivers are collected as input, along with properties of sewers and hydraulic structures. Moreover, ArcGIS applies the rainfall runoff (RR) module to divide the river and drainage basin and calculate the average rainfall with DEM characteristics. The Manning formula is used to calculate channel roughness along rivers and sewer systems. The White–Colebrook formula is used to calculate subcatchment surface roughness. In general, the higher the DEM spatial resolution, the higher the flooding simulation accuracy. However, increasing the resolution of the DEM is not necessary to improve the accuracy of flooding simulation such as flood inundation depth or extent, especially in the lowland areas. As shown in Figure 4, elevation at lowland areas of the Yilan River basin near the downstream boundary is less than 2 m. According to some test simulations with different spatial resolutions of DEM, it was found that using a spatial resolution of 40 m × 40 m DEM can obtain almost the same simulated results by 20 m × 20 m DEM for the flood inundation map in this study area. Therefore, the spatial resolution of 40 m × 40 m DEM is sufficiently accurate for flood inundation simulation in the study area, which can also result better computational efficiency, e.g., less CPU time.

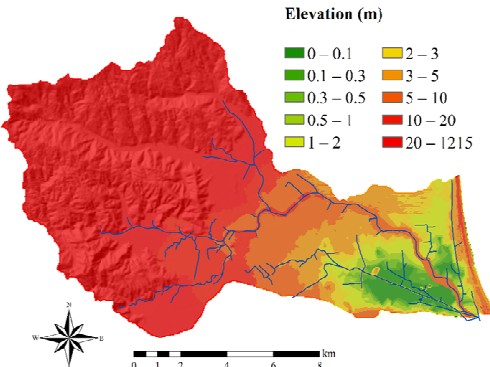

**Figure 4.** High-resolution elevation map of the Yilan River basin.

According to the typhoon warning information delivered from the Central Weather Bureau (CWB), Taiwan, the sequential 36 h rainfall data with most intensive rainfall is used for model calibration and forecasting. Moreover, obtaining from the CWB, the historical temporal datasets for model inputs are hourly-based hydrological data for each typhoon event, including rainfall intensity, tidal sea-level variations, etc. Then, to establish a stable and reasonable hydrodynamic initial condition, the warm-up period is needed. The warm-up time for the initial condition is 1 h at which we impose the same values of initial water depth and tidal water elevation at $t = 0$. Given the above information, the SOBEK model can be successfully applied to simulate the hydrodynamic characteristics of flood inundation.

### 3.4.2. Classification Step

For model efficiency, the $k$-means clustering algorithm is applied in grouping the flood inundation data into several clusters. First, flood inundation and non-inundation extents have to be identified. In this study, an extent with a flood inundation depth above 0.3 m is regarded as a flood inundation extent. The dominant clusters of flood gauging stations are then identified by minimizing the average Euclidean distance from the flood inundation data to the center of the cluster.

### 3.4.3. Point Forecasting Step

The point forecasting module for each flood gauging station is constructed in this step. Observed rainfall and water level are used as input for the point forecasting module. The point forecasting module can be written in a general form as

$$H'_c(t + \Delta t) = \text{SVM}[R(t), R(t-1), \ldots, R(t-n), H_c(t), H_c(t-1), \ldots, H_c(t-m)] \tag{12}$$

where $t$ is the current time, $\Delta t$ is the lead time period (from 1–3 h), $R$ is the rainfall, $H_c$ is the water level at a given flood gauging stations and $H'_c(t + \Delta t)$ is the point forecasted water level at time $t + \Delta t$.

The determination of the inputs and of appropriate lag lengths is crucial to the effectiveness of the proposed point forecasting module. Rainfall and water level are selected as input, while the lag lengths of input are determined by the grid search method.

### 3.4.4. Spatial Expansion Step

In this step, the spatial expansion module is developed to expand the forecasts from points to areas. First, the forecast of each flood gauging station is transformed to a flood inundation depth. Then, SVM is applied to expand the flood inundation depth from points to areas for each cluster. Inputs for developing the spatial expansion module include the forecasted flood inundation depths of flood gauging stations with grid coordinates, rainfall and nine flood causative factors. The spatial expansion module can be written as follows:

$$D'_n(t + \Delta t) = \text{SVM}[R(t), D'_c(t + \Delta t), I_n] \tag{13}$$

where $R$ is the rainfall, $D_c$ is the flood inundation depth of a given flood gauge station, $I_n$ is a given flood causative factor, and $D'_n(t + \Delta t)$ is the forecasted flood inundation depth of grid $n$ at time $t + \Delta t$.

### 3.5. Model Evaluation and Cross Validation

To evaluate model performance, six measures of error are employed to indicate the discrepancy between the observed and forecasted values: root mean square error (RMSE), mean absolute error (MAE), error of time to peak water level ($E_{Tp}$), error of peak water level ($E_{Wp}$), capture rate (CR) and coefficient of efficiency (CE). Smaller RMSE, MAE, $E_{Tp}$ and $E_{Wp}$ values indicate less significant errors between the observed and forecasted values, whereas higher CR value means better agreement of flood inundation extents between observed and forecasted values. The CE value is used to evaluate forecasting ability, with a CE value close to 1 representing high performance. In particular, RMSE and CE are selected as performance measures for the point forecasting module.

For the construction of the proposed model, the collected event-based data are usually separated into two sets: training and testing data. Training data are adopted to develop the proposed model, whereas testing data are used to evaluate the performance of the proposed model. Different selections of training and testing data do impact results, sometimes even leading to different conclusions. To evaluate the accuracy and robustness of the proposed model, a statistical technique called cross validation [47] is applied in this study. Cross validation is described in detail as follows. Each single typhoon event is chosen as the testing set in turn, while the remaining events are used as the training sets. Thus, for $N$ typhoons, each of the events is used to test the performance of the proposed model, and the test results and their performance measures are obtained. Then, performance conclusions for the proposed model are drawn on the basis of the averaged performance measures over all testing events.

## 4. Results and Discussion

### 4.1. Calibration and Validation of SOBEK

The observed flood inundation depths of the six water level stations on the Yilan River and Meifu drainage system are adopted in this study to calibrate and validate the SOBEK model. However, the selection of the parameter values of *CN*, *n* and *k* impacts the modeled values for flood inundation extent, depth and velocity. To determine the optimal parameter values for *CN*, *n* and *k*, these parameters are respectively varied from 39 to 98, from 0.015 to 0.035 and from 0.2 to 10. The determined optimal parameter values are similar to previous studies [6,48], which selected the same study area.

The performance of the SOBEK model at each water level station is shown in Table 2. As demonstrated in Figure 5, water level stations located downstream (i.e., Gamalan, Sijie, Dongjin and Zhuangwei) show good agreement between observed and simulated water levels. However, the $E_{Tp}$ and RMSE values at upstream water level stations (i.e., Yixing and Ximen) are worse than those at downstream water level stations. $E_{Tp}$ values below 2 h are acceptable, as are $E_{Wp}$ values below 10%. Meanwhile, CE values above 0.7 are acceptable [49]. Though the $E_{Wp}$ value at Dongjin is greater than 10%, the difference between observed and simulated peak water levels is only 0.5 m; therefore, we accept the errors of the simulated water levels as reasonable. Moreover, the simulated and observed flood inundation extents are in a good agreement (see Figure 6). The CR value of 78% indicates that the SOBEK model can accurately simulate the flood inundation extents. We accept the SOBEK model as an accurate and efficient way to simulate flood inundation in the Yilan River basin.

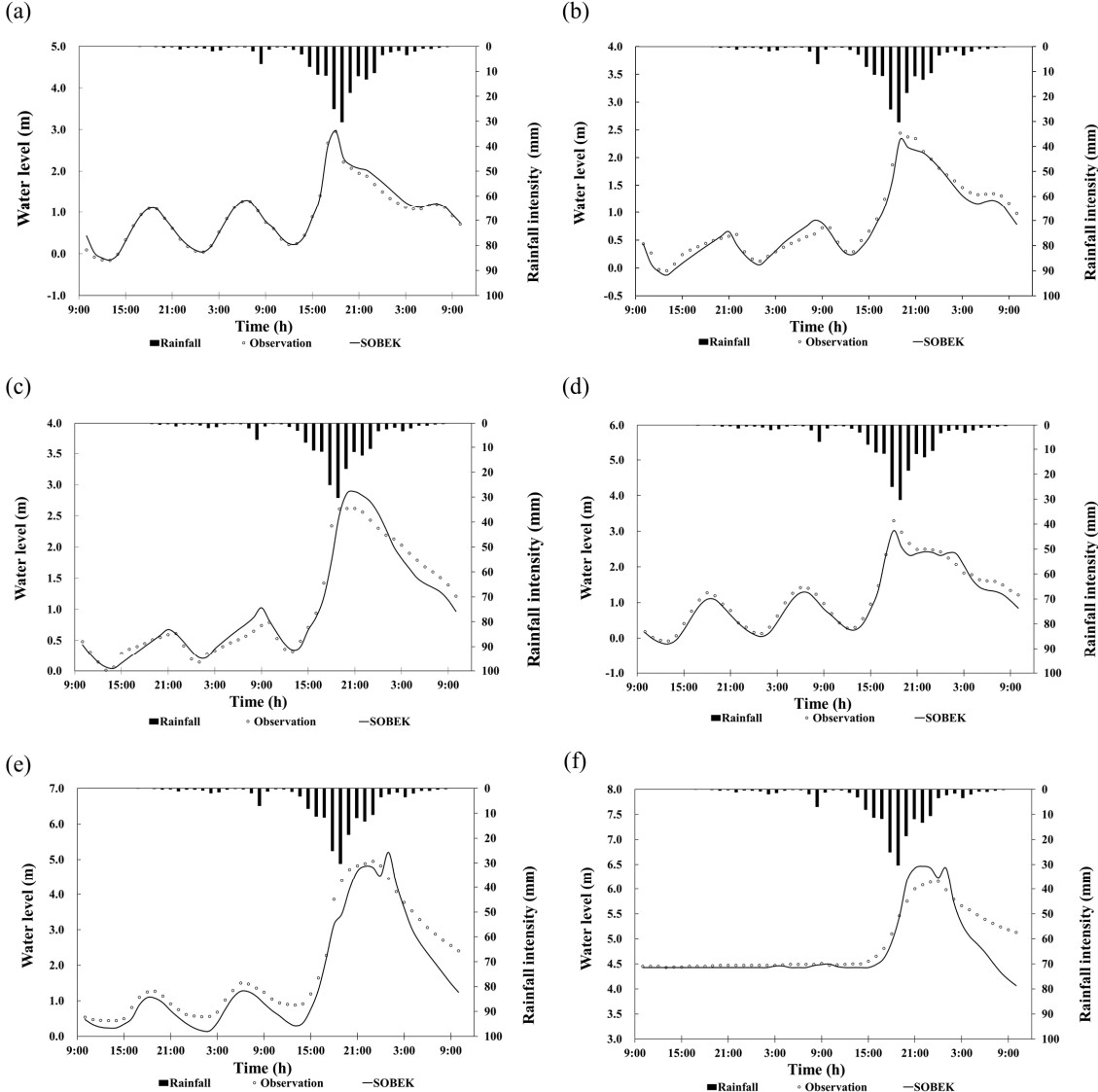

**Figure 5.** Comparison of observed water levels with simulated ones by SOBEK model for Typhoon Dujuan (from 27 September 2015 9:00 am to 28 September 2015 9:00 am) at (**a**) Gamalan, (**b**) Sijie, (**c**) Dongjin, (**d**) Zhuangwei, (**e**) Yixing, and (**f**) Ximen.

**Table 2.** $E_{Tp}$, $E_{Wp}$, CE and RMSE values for the water level: simulated results by SOBEK model.

| Water Level Station | $E_{Tp}$ (h) | $E_{Wp}$ (%) | CE | RMSE (m) |
|---|---|---|---|---|
| Gamalan | 0 | 0.34 | 0.98 | 0.09 |
| Sijie | 0 | 4.92 | 0.97 | 0.16 |
| Dongjin | 0 | 10.18 | 0.95 | 0.18 |
| Zhuangwei | 0 | 8.16 | 0.96 | 0.30 |
| Yixing | 2 | 5.07 | 0.90 | 0.76 |
| Ximen | 2 | 5.10 | 0.74 | 1.09 |

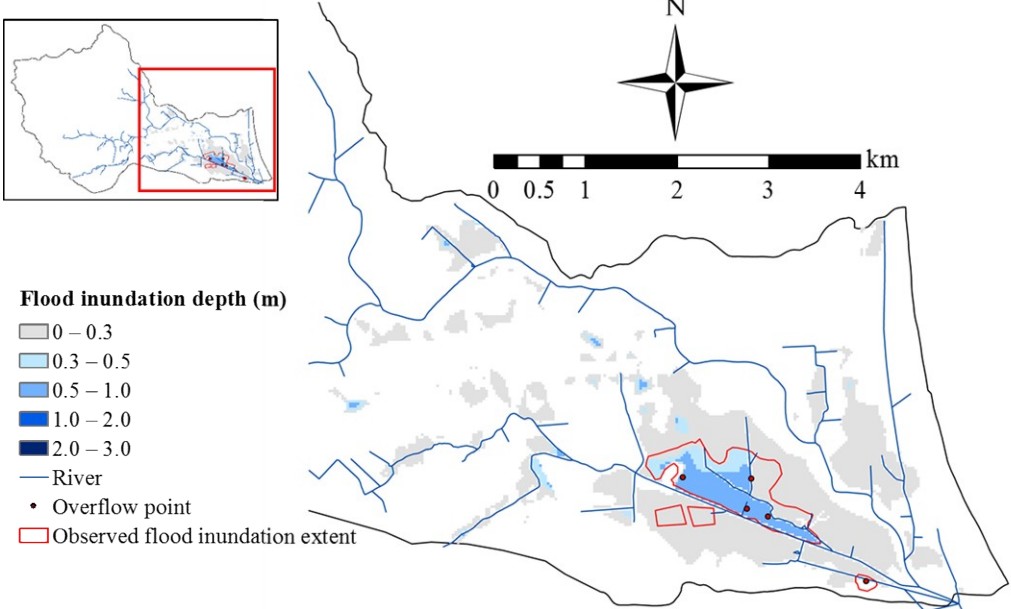

**Figure 6.** Comparison of simulated and observed flood inundation extents.

## 4.2. Identification of Clusters

In this subsection, the flood inundation depth within each cluster identified by the classification module is analyzed. The *k*-means clustering algorithm is applied to obtain the relative information of the flood inundation data in each grid. In this study, the optimal number of clusters is 10, determined by the grid-search method. The dominated clusters of flood gauging stations are listed in Table 3.

**Table 3.** Hydrologic descriptions and dominant clusters for flood gauging stations.

| Flood Gauging Station | Maximum Water Level (m) | Minimum Water Level (m) | Range of Water Levels (m) | Dominant Cluster |
|---|---|---|---|---|
| ISR2 | 5.18 | 2.37 | 2.81 | 3 |
| KXL1 | 2.73 | 0.22 | 2.51 | 6 |
| ISR3 | 6.57 | 4.33 | 2.24 | 1, 5 |
| ISR7 | 1.81 | 0.07 | 1.74 | 10 |
| GJL1 | 1.77 | 0.13 | 1.64 | 7, 8, 9 |

Figure 7 shows the results of the *k*-means clustering algorithm and the location of each flood gauging station. The main flood inundation extents are located between the Yilan River and Meifu drainage system, especially at low elevations (shown in Figures 4 and 7). Clusters 7–10, which contain areas located adjacent to the river show a distinct of flood inundation depth dynamic: peak flood inundation depths in Clusters 7–10 are above 1 m, while peak flood inundation depths in Clusters 1–6 are below 0.8 m. As shown in Figure 8, flood inundation depths are implicitly clustered according to their respective time series, including peak flood inundation depth and amplitude of flood inundation.

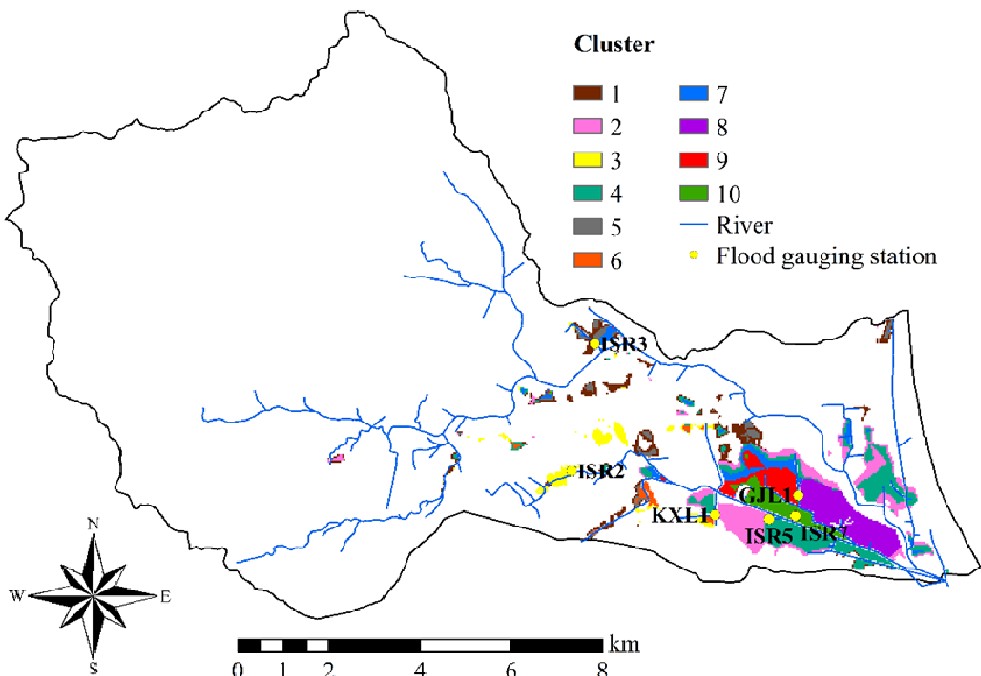

**Figure 7.** Data clustering results from the *k*-means clustering algorithm, along with locations of flood gauging stations.

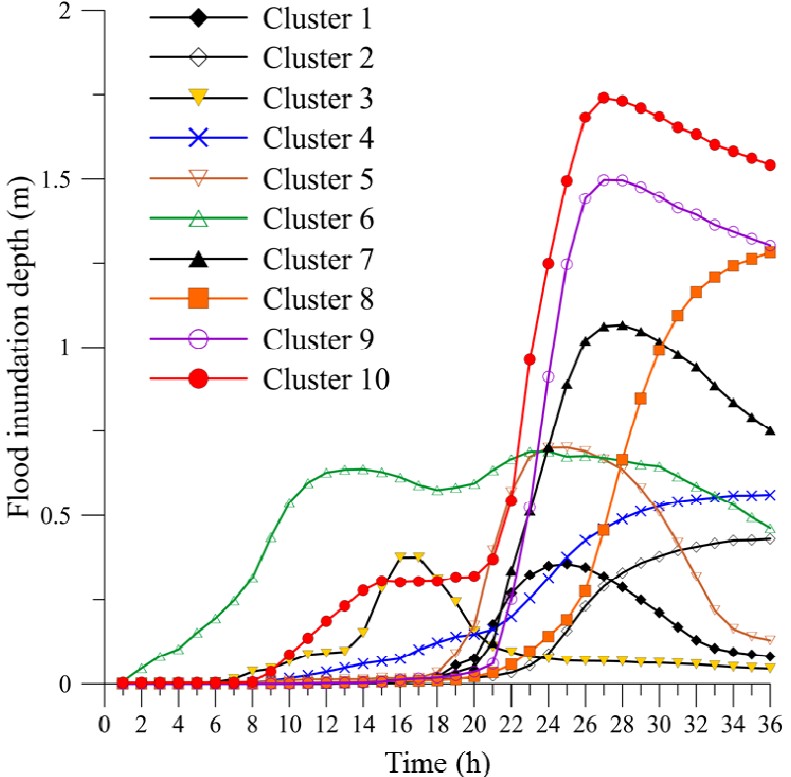

**Figure 8.** Average flood inundation depth of each cluster over time.

A narrower focus on the comparison between Clusters 7, 9 and 10, reveals that the flood inundation depth patterns are similar, with the highest peak in Cluster 10, followed by Cluster 9 and Cluster 7. The timing order of flood inundation is Cluster 10 first, then Cluster 9, and finally Cluster 7. These phenomena are consequences of the geographic distance from the river to each cluster's

geographic areas. For the same reason, Clusters 1 and 5 show similar dynamics, as do Clusters 2 and 4. We conclude that the *k*-means clustering algorithm effectively captures the spatiotemporal distribution of flood inundation depths.

### 4.3. Performance of the Point Forecasting Module

Observed rainfall and water level are the inputs for point forecasting. Based on the grid search method, the lag lengths for rainfall and water level are both determined to be 1, representing 1 h lead time. Current rainfall and water level are both determined as input for 2 to 3 h lead time.

Figure 9 presents the point forecasting results for each flood gauging station for 1–3 h lead time. Most of the water level forecasts from the point forecasting module are in good agreement with the observed data. However, the peak flood inundation depths at ISR2, KXL1, ISR3 and GJL1 are underforecasted for Typhoons Nonmadol and Sinlaku, especially for 2 to 3 h lead time. We therefore employed RMSE and CE to clearly and objectively compare the discrepancies between observed and forecasted water levels. As shown in Table 4, with increasing forecast lead time the RMSE values increase (i.e., worsen) and the CE values decrease (i.e., worsen).

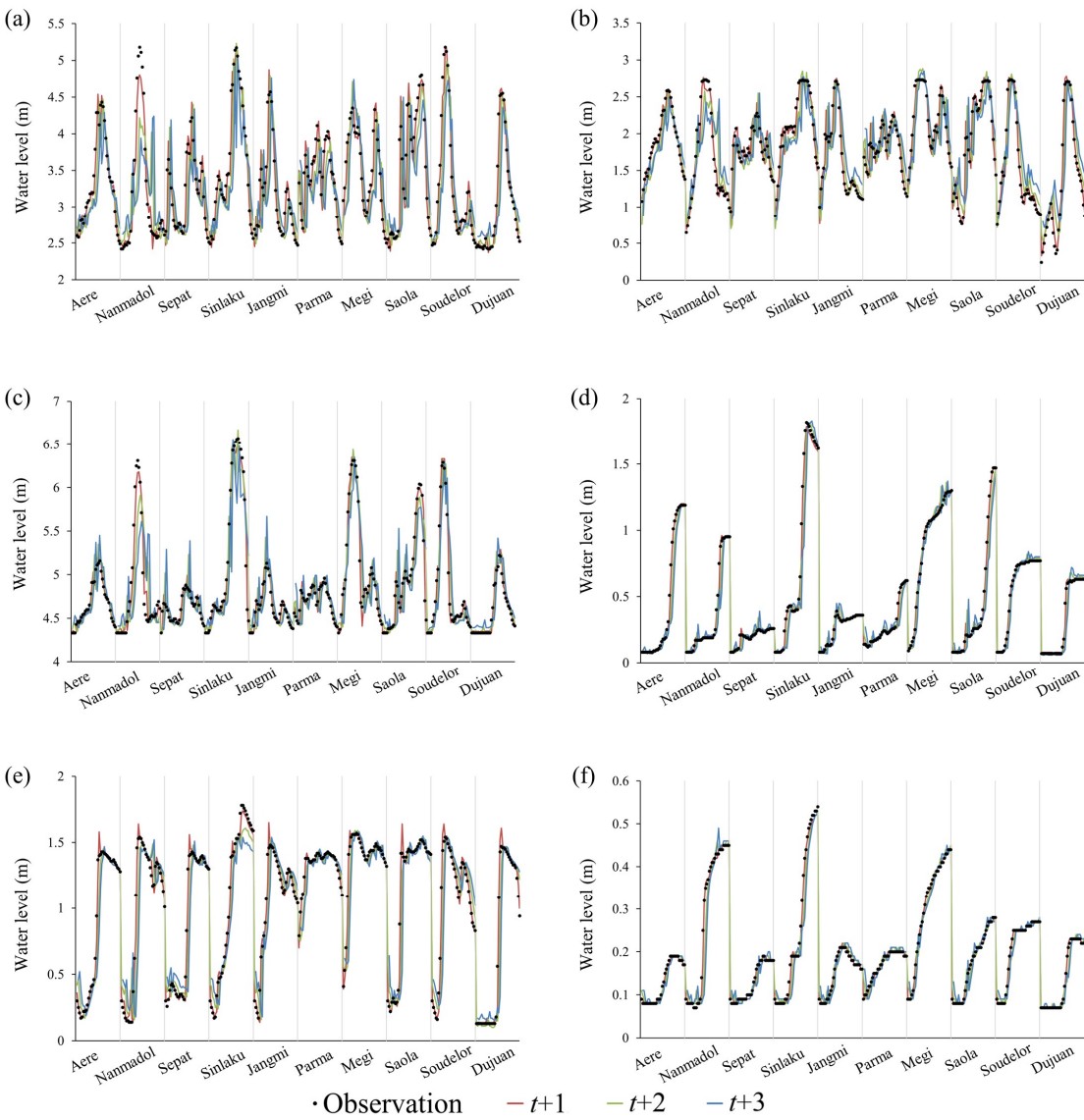

**Figure 9.** Comparison of observed and forecasted water levels for 1–3 h lead time at (**a**) ISR2, (**b**) KXL1, (**c**) ISR3, (**d**) ISR7, (**e**) GJL1, and (**f**) ISR5.

**Table 4.** RMSE and CE values for the point forecasting module for 1–3 h lead time forecasting.

| Lead Time (h) | ISR2 | KXL1 | ISR3 | ISR7 | GJL1 | ISR5 |
|---|---|---|---|---|---|---|
| RMSE (m) | | | | | | |
| $t+1$ | 0.19 | 0.12 | 0.09 | 0.03 | 0.07 | 0.01 |
| $t+2$ | 0.41 | 0.26 | 0.19 | 0.07 | 0.17 | 0.02 |
| $t+3$ | 0.55 | 0.36 | 0.27 | 0.10 | 0.24 | 0.02 |
| CE | | | | | | |
| $t+1$ | 0.90 | 0.92 | 0.93 | 0.98 | 0.97 | 0.99 |
| $t+2$ | 0.57 | 0.69 | 0.67 | 0.91 | 0.81 | 0.94 |
| $t+3$ | 0.26 | 0.37 | 0.29 | 0.79 | 0.56 | 0.88 |

A comparison of Tables 3 and 4 clearly shows that with decreasing water level amplitude, RMSE values decrease (i.e., improve) while CE values increase (i.e., improve), with the exception of GJL1. The dynamics at GJL1 could be caused by the water level pattern as well as underforecasting. The water level pattern at GJL1 is fluctuant like ISR2, KXL1 and ISR3, and notably the peak flood inundation depths are underforecast at all these locations. Moreover, the RMSE and CE values at ISR2 increase from 0.19 m to 0.55 m and decrease from 0.9 to 0.26, respectively. The proposed model is relatively low performance at ISR2 and the RMSE value increases 65% from 1–3 h ahead time. Despite this, the performance deterioration for the proposed model is slower than that obtained by Lin et al. [27] (69%). Based on the aforementioned results, we conclude that the proposed point forecasting module yields sufficiently high forecast accuracy for most gauging stations.

*4.4. Performance of the Spatial Expansion Module*

The performance measures for the proposed spatial expansion module are presented in Table 5. With increasing forecast lead time, RMSE and MAE values increase (i.e., worsen). As the forecast lead time increases, the more noises are included between inputs and target output and hence the more errors in the forecasts.

**Table 5.** RMSE and MAE values for the spatial expansion module for 1–3 h lead time forecasting.

| Lead Time (h) | Cluster | | | | | | | | | |
|---|---|---|---|---|---|---|---|---|---|---|
| | 1 | 2 | 3 | 4 | 5 | 6 | 7 | 8 | 9 | 10 |
| RMSE (m) | | | | | | | | | | |
| $t+1$ | 0.07 | 0.09 | 0.08 | 0.13 | 0.14 | 0.20 | 0.20 | 0.13 | 0.35 | 0.11 |
| $t+2$ | 0.07 | 0.08 | 0.08 | 0.13 | 0.15 | 0.18 | 0.19 | 0.12 | 0.32 | 0.13 |
| $t+3$ | 0.07 | 0.08 | 0.08 | 0.13 | 0.14 | 0.18 | 0.20 | 0.12 | 0.31 | 0.16 |
| MAE (m) | | | | | | | | | | |
| $t+1$ | 0.05 | 0.05 | 0.05 | 0.09 | 0.09 | 0.15 | 0.14 | 0.10 | 0.26 | 0.07 |
| $t+2$ | 0.05 | 0.05 | 0.05 | 0.08 | 0.10 | 0.13 | 0.13 | 0.09 | 0.24 | 0.09 |
| $t+3$ | 0.04 | 0.05 | 0.05 | 0.09 | 0.09 | 0.13 | 0.13 | 0.09 | 0.24 | 0.11 |

RMSE and MAE values across clusters are also apparently influenced by maximum flood inundation depth. As shown in Figure 8, RMSE and MAE values increase with increasing number of clusters (i.e., with increasing the maximum flood inundation depth). As can be seen in Table 5, maximum flood inundation depths in Clusters 1–5 remain below 0.56 m, and in these same clusters RMSE values remain below 0.16 m, and MAE values remain below 0.11 m.

In addition, the more accurately the model forecasts the water level of the cluster's flood gauging station, the more accurately it forecasts the cluster's flood inundation depths. Due to excellent model performance at ISR7, Cluster 10's RMSE and MAE values are lower than 0.16 m and 0.11 m, respectively. In contrast, due to relatively poor model performance at ISR2 and KXL1, the spatial expansion module in Clusters 6–9 performs worse. Despite this, most of the results confirm that the proposed spatial expansion module effectively forecasts flood inundation depths.

To highlight the forecasting performance of the proposed spatial expansion module, the recent Typhoon Dujuan (2015) is taken as an example. Figure 10 presents the flood inundation data versus corresponding forecasts for flood inundation depth at 1–3 h lead time for Typhoon Dujuan. The forecasted flood inundation depths are generally in good agreement with the flood inundation data. As above, with increasing forecast lead time, the difference between observed values and forecasted values increases. Figure 11 shows MAE performance across the flood inundation map at 1–3 h lead time for Typhoon Dujuan. High performance is indicated by MAE values between 0 and 0.1 m; at 1–3 h lead time we observe high performance covering 85.2%, 84.2% and 83.4% of the map area, respectively. Meanwhile, MAE values exceeding 0.5 m rarely occur. Compared to Chang et al. [18], the proposed model obtains higher percentage of MAE values between 0 and 0.1 m. It indicates that the proposed model yields the accurate flood inundation map.

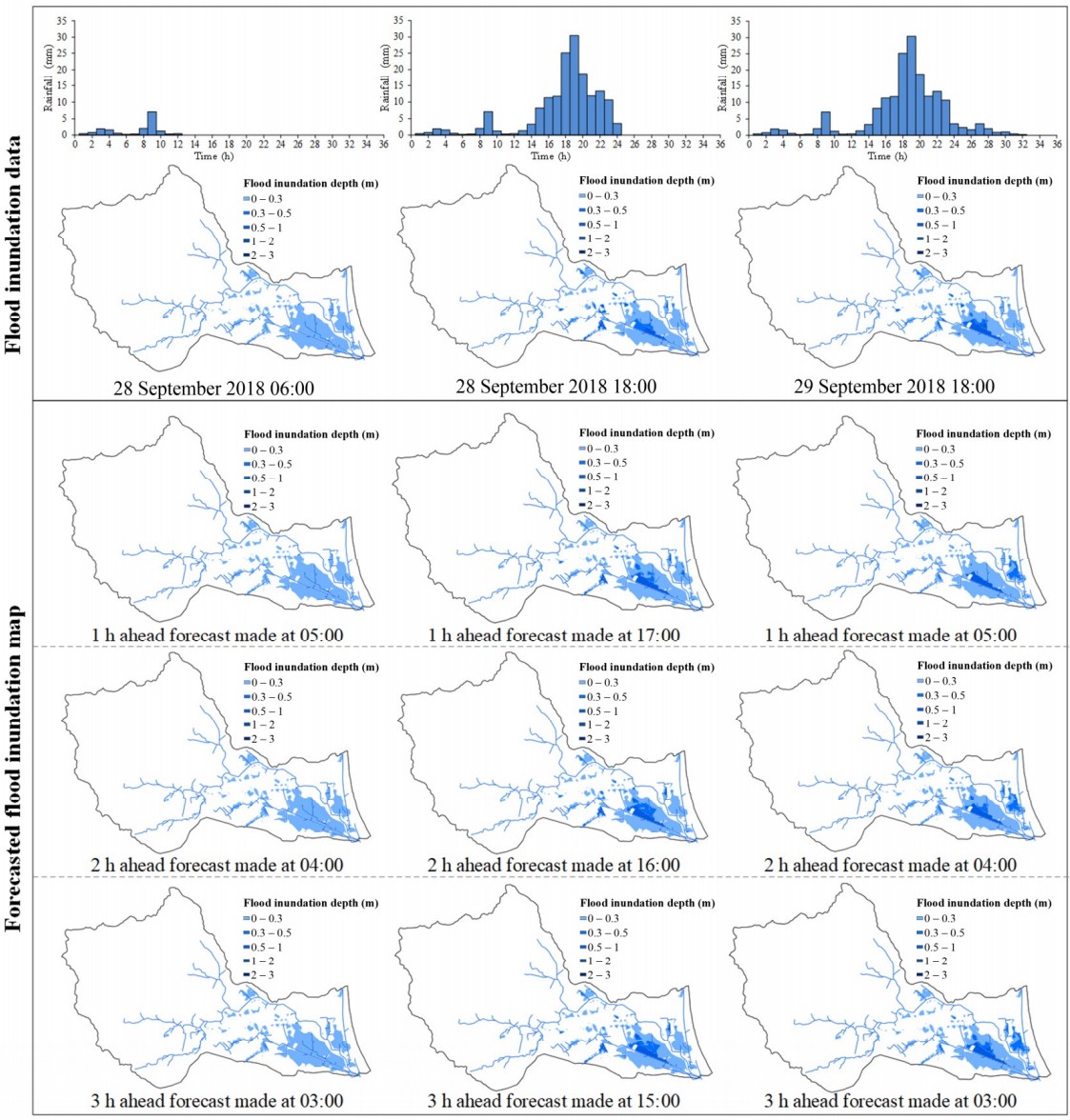

**Figure 10.** Comparison of flood inundation data with forecasts obtained from the proposed model for Typhoon Dujuan.

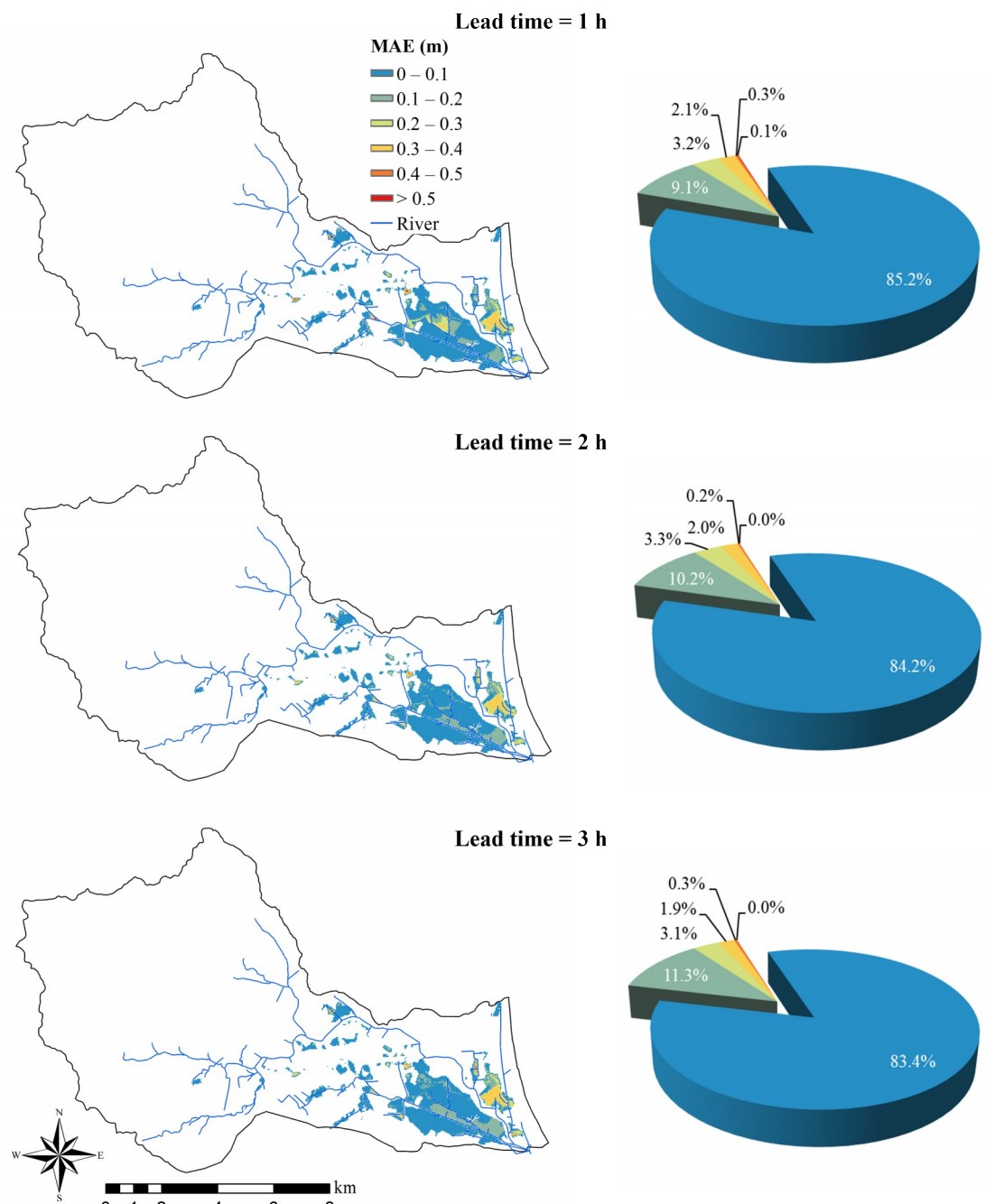

**Figure 11.** Distribution of MAE value with respect to the proposed model during Typhoon Dujuan.

The aforementioned results indicate that the proposed model accurately forecasts not only flood inundation depth, but also areal flooding extent. The proposed spatial expansion module accurately and reliably produces flood inundation forecast maps, providing information to help disaster mitigation and reduce loss of life and property.

## 5. Conclusions

Accurate forecasts for flood inundation depth and extent play a crucial role in flood early warning systems. For this purpose, an accurate and efficient forecasting model is proposed for producing flood inundation forecast maps.

Based on various measures of error, we find that the SOBEK model accurately simulates flood inundation depths. Flood inundation depth at most locations shows model errors less than 0.1 m, making the proposed flood inundation forecasting model suitable for flood inundation forecasts in typhoon.

The point forecasting module accurately forecasts water level for 1–3 h lead time for most flood gauging stations. Finally, based on the strong performance of the spatial expansion module, we conclude that the proposed model can provide acceptable flood inundation forecast maps for 1–3 h lead time. The accurate forecast maps of inundation depth and extent are expected to be useful for flood early warning systems. They also can be helpful for decision makers in handling disaster emergency response, reducing human casualties and property damage.

Due to some error of simulated flood inundation depths resulted from hydrodynamic modeling, it may cause poor performance in the flood inundation forecasting model. In the future, the proposed forecasting model can use real-time monitoring data and hence produce more accurate flood inundation forecast maps. Meanwhile, novel clustering algorithms and ANNs probably may improve the model performance.

**Author Contributions:** M.-J.C. developed the ANN model and took the lead in writing the manuscript; H.-K.C. and Y.-C.C. performed the hydrodynamic simulation and analyzed the data; P.-A.C. contributed to the analysis of the results and to the writing of the manuscript; G.-F.L. supervised the research work; J.-S.L. and Y.-C.T. provided guidance and suggestion to the manuscript.

**Funding:** This paper is based on work partially funded by Ministry of Science and Technology, Taiwan, under Grants MOST 107-2119-M-002-019 and MOST 107-2628-M-002-016.

**Acknowledgments:** The authors gratefully acknowledge the support by providing data from the Central Weather Bureau and the Water Resources Agency, Taiwan. The authors also thank the Hydrotech Research Institute of National Taiwan University for using facilities and technique support. Finally, we would like to thank the editors and reviewers for their constructive suggestions that greatly improved the manuscript.

**Conflicts of Interest:** The authors declare no conflict of interest.

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
