# Peer review of "A Support Vector Machine Forecasting Model for Typhoon Flood Inundation Mapping and Early Flood Warning Systems"

_water, doi:10.3390/w10121734_

Round 1

Reviewer 1 Report

The manuscript presents an integrated forecasting system for the flood inundation mapping. The system includes a hydrodynamic simulation model, a k-means clustering algorithm and an expansion module to forecast the flood extension.

The article is well written and the contents are interesting for the readers of the journal. At the best of my knowledge, the presented complete system is original. I suggest to accept the article in the present form.

Author Response

The referee’s comment is much appreciated. 

Reviewer 2 Report

  The paper presents a interesting flood forecasting method incorporating a multianalysis framework. The work could have a significant merit and it looks that a massive work has been done by the authors. I would like to propose it for publication after major revisions. Below the authors can be found my recommendations for revisions:

Introduction

I would suggest that the introduction section should be most to the point. It looks like the authors jump from modeling technique without saying specifically the capability of its framework. I would suggest to work again the introduction section between lines 54-100 by representing a more clear message across the different models. I would also to suggest to highligth the uncertainty which is ubiquitous in the phenomenon (Dimitriadis et al. 2016).

Line 119:Could you clarify what is SPI, TWI?

Line 210: Clasification of what? 

Table 1: I do not believe that the row of inundation extent is part of hydrological data.

Please use units in every equations (Lines 145 etc)

Chapter 3.2: Clarifications is needed to this section. It not clear specifically the phrase In this study, the similar flood inundation data are grouped in the same cluster for boosting forecast  accuracy.

Chapter 3.3 Again here very confused chapter with invisible equations. Not clear the usability of SVM in our case.

Chapter 4.1: I would rephrase the title as Calibration and validation of hydrological and hydraulic model. The SOBEK is just as a tool

Figure 6: I would suggest to reclasify the categories of flood extents with different colours and could you also zoom in the area of interest? Please minimize the extent of the legend.

Conclusion: Poor section with a such rish work. Limitations of the proposed could also represent here, future improvements further discussion issues.

References

Dimitriadis, P., Tegos, A., Oikonomou, A., Pagana, V., Koukouvinos, A., Mamassis, N., ... & Efstratiadis, A. (2016). Comparative evaluation of 1D and quasi-2D hydraulic models based on benchmark and real-world applications for uncertainty assessment in flood mapping. Journal of Hydrology534, 478-492.

Author Response

(The authors gave the same response as above.)

Reviewer 3 Report

I enjoyed reading your manuscript. The methods are well presented, and the experiments are well designed. The results provide very valuable information from an operational forecasting point of view. I only have minor comments listed below. I think the manuscript is ready to be published on Water.

Lines : 68 and 69. This is a strong statement. Is this an opinion of the authors or has this statement been corroborated by studies?

 Some equations don’t look fine in the final pdf. Look blurry. Please fix of equations of lines 148, 155, 161, 186, 187, 191, 192, 194, 195, all equations of subsection 3.3

Please indicate in subsection 3.3 that this section provides the SVM background required for the Point forecasting step.

Please avoid blank lines like in 267 and 268. Or move line 269 to page 13

Author Response

(The authors gave the same response as above.)

Reviewer 4 Report

This paper is mainly about the flood forecasting and inundation mapping. During the recent years, and with the increase in frequency of events like hurricanes and typhoons, this field of research is among the hot-topics. The paper is almost well-organized and well-written, while it needs more revisions before final publication.

Some improvements regarding the English writing is needed. Try to use shorter sentences with the correct tenses.

The abstract needs more results with more details.

You need to explain clearly how the DEM spatial resolution affect your computation accuracy?

Figure 1 caption is incomplete and needs revision.

Why did you choose 36 hours? There is an event which is less than 36 hours (35h)

Clearly explain how temporal and spatial resolution of your datasets can affect your models’ output?

Figure 2 contains some images that are not very informative and it is just copy and paste. You need to remove them from the main text or at least provide informative information on their importance and their role in your model.

Lines 155, 161, 186, and equation number 7, 8, and … are more like image. You need to type them directly in your text.

Explain more about the model calibration process. What is your train data and test data?

Explain how you run the model? Did you set the warm-up period for the model?

You need to compare your results with other studies in more details in the discussion section. Try to add some more recent references and studies to your paper. Also, try to use references from different authors that used some other hydrologic/hydrodynamic models.

It is very important to show how location of rainfall stations can affect the accuracy of your data as input datasets and how it changes the models output?

Author Response

(The authors gave the same response as above.)

Round 2

Reviewer 2 Report

Dear Editor,

My review comments and concerns have fully adressed and therefore I recommend for publishing.

Reviewer 4 Report

fine for final publication